# Systematic review and meta-analysis on juvenile primary spontaneous pneumothorax: Conservative or surgical approach first?

Chun-Shan Hung[1], Yang-Ching Chen[2,3], Ten-Fang Yang[4,5], Fu-Huan Huang[4,6]*

1 Department of Medical Education, Changhua Christian Hospital, Changhua City, Taiwan, 2 Department of Family Medicine, Taipei Medical University Hospital, Taipei City, Taiwan, 3 Department of Family Medicine, School of medicine, College of medicine, Taipei Medical University, Taipei City, Taiwan, 4 Department of Biological Science and Technology, National Chiao-Tung University, Hsinchu City, Taiwan, 5 Graduate Institute of Biomedical Informatics, Taipei Medical University and Hospital, Taipei City, Taiwan, 6 Division of Pediatric Surgery, Department of Surgery, Taipei Medical University Hospital, Taipei City, Taiwan

* 191009@h.tmu.edu.tw

## Abstract

### Background

Primary spontaneous pneumothorax (PSP) prevalence is typically higher in juvenile patients than in adults. We aimed to evaluate the optimal treatment for primary spontaneous pneumothorax and its efficacy and safety in juveniles.

### Materials and methods

We searched PubMed, Embase, and Cochrane databases for eligible studies published from database inception to October 10, 2020, and conducted a systematic review and meta-analysis according to Preferred Reporting Items for Systematic Reviews and Meta-Analyses guidelines. The primary and secondary outcomes were recurrence rate and hospital stay length, respectively. Odds ratios (OR) and mean differences were used for quantitatively analyzing binary and continuous outcomes, respectively. In total, nine retrospective studies with 1,452 juvenile patients (aged <21) were included for the quantitative analysis. The surgical approach led to a lower recurrence rate than did conservative approaches (OR: 1.95, 95% confidence interval: 1.15–3.32). Moreover, the recurrence rate was low in patients who underwent conservative treatment first and received surgery later.

### Conclusions

Surgical approach for first-line management might have a greater effect on recurrence prevention than do conservative approaches. An upfront surgery might be an optimal choice for juvenile primary spontaneous pneumothorax.

**Data Availability Statement:** All relevant data are within the paper and its Supporting information files.

**Funding:** The authors received no specific funding for this work.

**Competing interests:** The authors have declared that no competing interests exist.

## Introduction

Spontaneous pneumothorax can be classified as primary spontaneous pneumothorax (PSP) and secondary spontaneous pneumothorax. PSP, defined as pneumothorax without underlying lung disease, occurs frequently in men with a tall and thin body habitus. PSP is rarely encountered in young children, with an incidence of 3.4 per 100,000 children aged <18 years [1]. Because relatively few studies have focused on children, the clinical characteristics and outcomes of PSP in this age group warrant further investigation.

Four general approaches are used for the initial management of patients with PSP: (A) observation only, (B) oxygen supplementation, (C) drainage through needle aspiration or chest tube placement, and (D) immediate operation, most commonly video-assisted thoracoscopic surgery (VATS). Certain groups of experts advocate early surgical approach for definite disease resolution [2, 3]. However, some professionals reserve surgery for persistent air leaks or recurrent PSP [4, 5]. Furthermore, no evidence-based pediatric-specific guidelines for spontaneous pneumothorax management exist and guidelines for adult patients have been applied to the pediatric population [6, 7].

Some patients with PSP experienced recurrence even after surgery. Although PSP recurrence pathophysiology remains uncertain, it appears to be age related. A review of a Taiwanese nationwide database [8] revealed an age-stratified incidence and 1-year PSP recurrence rate that has been demonstrated to decrease with age, especially greater than 21. Moreover, the recurrence rates of PSP after conservative and operation management were 23.74% and 14.14% respectively.

Due to the inconsistency and lack of evidence, substantial variations existed in the approaches used for the initial PSP management. Therefore, we conducted a systematic review and meta-analysis of the up-to-date literature to investigate the effectiveness of the surgical approach compared with conservative approaches (such as supplemental oxygen, pigtail, and chest tube) as the initial treatment for PSP in juvenile patients (aged<21).

## Materials and methods

This prospective systematic review was initiated on September 29, 2020, and the study protocol was designed in advance. The primary design was registered on PROSPERO (CRD42020212606). The study group included a pediatric surgeon and researcher experienced in systematic review and meta-analysis. This study conformed to the Preferred Reporting Items for Systematic Reviews and Meta-Analyses (PRISMA) guidelines for evidence selection, quality assessment, evidence synthesis, and research reporting [9].

### Eligibility criteria

Studies were included if they (1) were randomized controlled trials or prospective or retrospective studies with recurrence rate outcome, (2) included a juvenile population (age < 21 years) with PSP diagnosis, and (3) compared the surgical and conservative approaches for PSP treatment. No language criterion was applied to the studies. Case reports, reviews, commentaries, and conference abstracts were excluded.

### Search strategy and study selection

The keywords were combined using appropriate Boolean operators, and a primary search strategy was developed without limitations regarding language and published data. The primary search strategy involved a PubMed search, which was adapted to Embase and Cochrane database (S1 Table). The final search was completed on October 10, 2020. The reference lists of

retrieved articles were reviewed for undetected relevant studies. The literature search and all retrieved studies were independently reviewed by two authors (F.H.H. and C.S.H.).

## Data extraction

In total, 210 duplications were eliminated, and the titles and abstracts of the remaining studies were screened to remove irrelevant articles. Two reviewers (F.H.H. and C.S.H.) independently extracted data from all the included studies. Disagreements between reviewers were resolved by another author (Y.C.C). Data regarding the sample size, recurrence rate, and length of hospital stay after different interventions (including conservative and surgery approaches) were extracted.

## Quality assessment

Each study's methodological quality was evaluated using the Risk Of Bias In Non-randomized Studies (ROBINS-I) tool [10], which assesses the quality of nonrandomized case–control studies. The scoring system contains three major domains (preintervention, intervention, and postintervention) and overall risk of biased judgments (low, moderate, severe, and critical). Results were independently reviewed and discussed by two authors (F.H.H. and C.S.H.). The ROBINS-I quality assessment is presented in S2 Table.

## Primary and secondary outcomes

The primary outcome was PSP recurrence rate after PSP management with either conservative or surgical approach. Odds ratios (ORs) were used for determining the effect size. The secondary outcome was the length of hospital stay and mean differences were used for analysis.

## Statistical analysis

The OR and mean differences with 95% confidence intervals (CIs) were estimated for binary data. Heterogeneity among studies was quantified based on their $I^2$ value: an $I^2$ of >75%, >50%, and <25% was considered to indicate high, moderate, and low heterogeneity, respectively [11]. A random effect model was applied for all analyses. Two-sided P values < 0.05 were considered statistically significant. Review Manager (version 5.3; The Nordic Cochrane Centre, The Cochrane Collaboration, 2014) and R studio for Microsoft Windows (version 1.2.5001) were used for statistical analysis.

# Results

## Study characteristics

Fig 1 illustrates the study screening and selection process. The initial screen yielded 2,639 citations. After the exclusion of duplicate articles (n = 210), 2,429 citation records remained. Thereafter, on the basis of title and abstract screening of these records, 2,403 ineligible studies were excluded. Full texts of 26 articles were assessed to determine their eligibility, and 17 citation records (1 case report, 6 conference reports, 4 not involving PSP, and 6 with insufficient data to compute an effect size) were excluded. No randomized controlled trials were found during this literature search. Finally, nine studies [3, 4, 6, 12–17] were included in the meta-analysis.

The nine studies were retrospective studies on 1,452 patients, enrolled over 1964–2015, conducted in Israel (one study), the United States (five studies), Hong Kong (one study), Australia (one study), and France (one study). Four studies [3, 4, 14, 15] included patients aged <18 years, whereas the remaining studies applied a varied range of patient age: 10–18 [16],

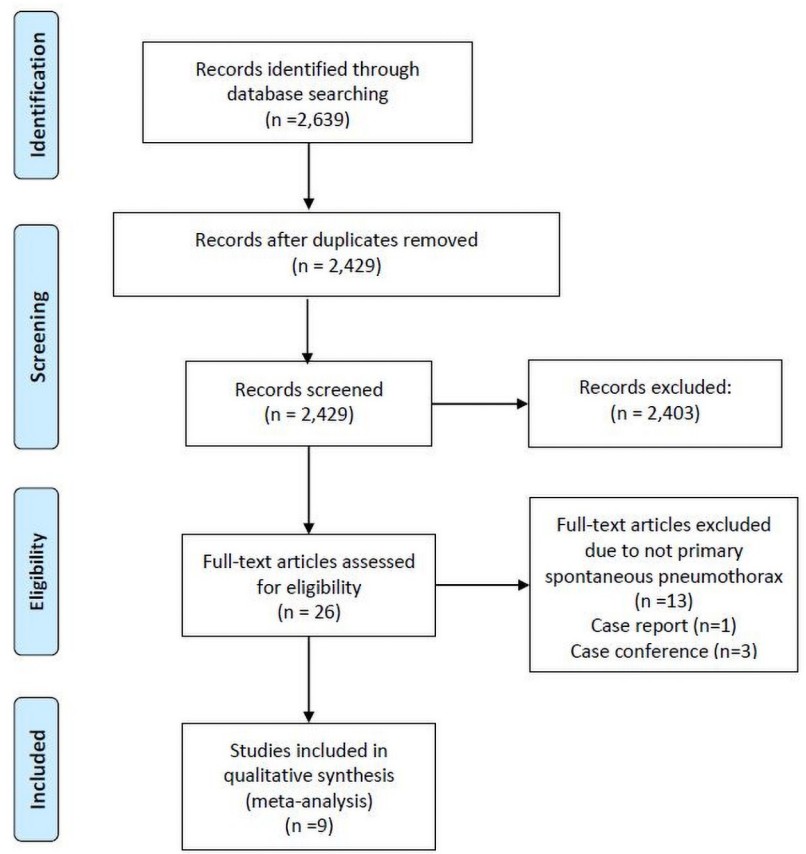

**Fig 1. PRISMA flowchart.**

11–17 [17], 10–19 [6], 5–20 [12], and 12–21 [11] years. Baseline patient characteristics of all nine studies are listed in Table 1.

## Conservative approach versus surgical approach as first-line management

Analyses were performed to examine the different recurrence rates between conservative and primary surgical approaches as the first-line management of PSP (Fig 2). The results showed that the surgical approach has a lower recurrence rate than the conservative approach (OR = 1.95, 95% CI = 1.15–3.32, P = 0.01). The total heterogeneity among the studies was non-significant ($I^2$ = 22%, P = 0.22). The recurrence incidences are 42.12% and 26.95% in the conservative and primary surgical approach group, respectively.

A subgroup analysis was performed to investigate whether different conservative approaches (observation only, oxygen support, pig tail catheter, and chest tube insertion) affect the recurrence rate: Observation alone versus primary surgical approach was analyzed in two trials [12, 13], involving 41 patients who were under observation only and 24 who underwent primary surgery, with no significant differences between the two groups (OR = 1.79, 95% CI = 0.51–6.26, P = 0.37). Heterogeneity among the studies showed no significant differences ($I^2$ = 0%, P = 0.35).

Oxygen supplement versus primary surgical approach was analyzed in two studies [3, 18], involving 43 patients who underwent oxygen supplementation only and 20 who underwent

**Table 1. Study characteristics.**

| Author (year) | Area | Study type | Total number of patients (n) | Patient numbers classified by treatments(n) | Mean age (years) | Study period | Follow up (years) | Quality |
|---|---|---|---|---|---|---|---|---|
| **Davis (1993) [17]** | Australia | R | 12 (6 males, 50%) | Thoracotomy:6 Chest tube:6 | 14.8 | 1964–1989 | N/A | 1 low risk |
| | | | | | | | | 5 moderate |
| | | | | | | | | 1 serious |
| **Qureshi (2005) [4]** | United States | R | 43 | Primary VATS:14 Chest tube:37 | 15.9 (0.35) | 1991–2003 | N/A | 1 low risk |
| | | | | | | | | 5 moderate |
| | | | | | | | | 1 serious |
| **Hui (2006) [16]** | Hong Kong | R | 63 (55 males, 87%) | Surgical treatment:15 Chest tube:48 | 16.5 (1.3) | 1997–2003 | 0.4 to 6.9 | 1 low risk |
| | | | | | | | | 6 moderate |
| **Nathan (2010) [15]** | France | R | 25 (17 males, 68%) | Primary VATS:12 | 14.2 (1.9) | 2000–2007 | 3.7 ± 2 | 1 low risk |
| | | | | Chest tube:13 | | | | 6 moderate |
| **Seguier-Lipszyc (2011) [3]** | Israel | R | 46 (40 males, 87%) | Primary VATS:10 | 16.2 | 1999–2009 | Primary VATS: 5.1 ± 3.3 | 1 low risk |
| | | | | Oxygen alone:18 | | | Oxygen alone: 5.4 ± 3.5 | 6 moderate |
| | | | | Chest tube:18 | | | Chest tube: 5.5 ± 2.6 | |
| **Lopez (2014) [14]** | USA | R | 96 (76 males, 79%) | Primary VATS:10 | 16.6 | 2005–2011 | 1 | 6 moderate |
| | | | | Oxygen alone:25 | | | | 1 serious |
| | | | | Pig tail:24 | | | | |
| | | | | Chest tube:49 | | | | |
| **Williams (2017) [6]** | USA | R | 1,040 (854 males, 82.1%) | VATS:207 | 15.7 (1.7) | 2010–2014 | N/A | 1 low risk |
| | | | | No intervention:336 | | | | 6 moderate |
| | | | | Chest tube:497 | | | | |
| **Soler (2018) [13]** | USA | R | 81 (61 males, 75%) | Primary VATS:14 | 17.1 (2.6) | 2002–2014 | N/A | 1 low risk |
| | | | | Observation only:33 | | | | 5 moderate |
| | | | | Pig tail:15 | | | | 1 serious |
| | | | | Chest tube:18 | | | | |
| **Williams (2018) [12]** | USA | R | 46 (41 males, 89%) | Primary VATS:10 | 16.1 (1.3) | 2007–2015 | N/A | 1 low risk |
| | | | | Observation only:8 | | | | 5 moderate |
| | | | | Chest tube:28 | | | | 1 serious |

Abbreviations: R, retrospective; VATS, video-assisted thoracoscopic surgery.

primary surgery, with no statistical differences between the two groups (OR = 2.06, 95% CI = 0.43–9.90, P = 0.37). Heterogeneity among the studies did not reveal significant differences ($I^2$ = 45%, P = 0.18).

Pig tail versus primary surgical approach was reported in two studies [13, 18], including 39 patients with pig tail insertion and 24 who underwent primary surgery, but the difference was nonsignificant (OR = 2.02, 95% CI, 0.64–6.42, P = 0.23). Heterogeneity was not significant among the studies ($I^2$ = 0%, P = 0.34).

Chest tube versus primary surgical approach were analyzed in seven studies [3, 4, 12–15, 17], involving 292 patients with chest tube insertion and 73 who underwent surgery. The recurrence number in the two groups did not differ significantly (OR = 2.07, 95% CI = 0.83–5.16, P = 0.12), and no significant heterogeneity existed among the groups ($I^2$ = 49%, P = 0.07).

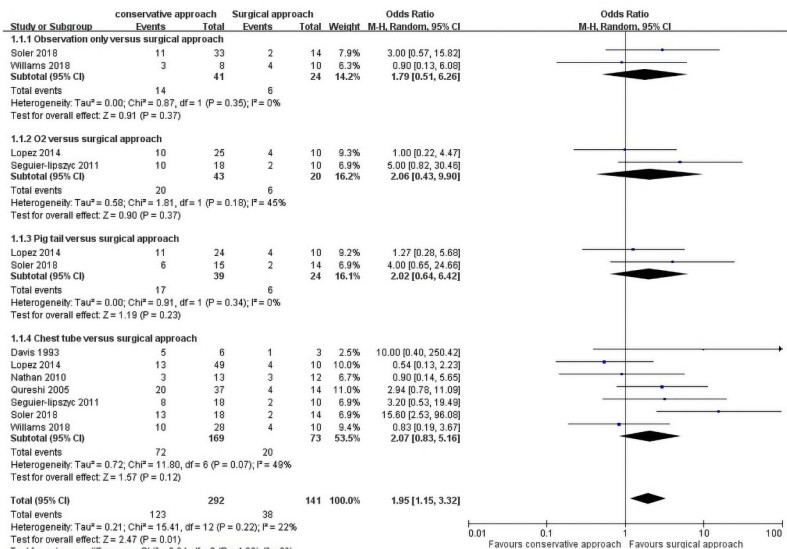

**Fig 2. Recurrence rate of conservative approaches versus surgical approach as first-line management.**

## Conservative approaches only versus surgical approach after conservative approaches

The outcomes are shown Fig 3, with three studies included [12, 16, 18]. In total, 160 patients underwent only conservative intervention, and 64 patients underwent surgery after the conservative intervention. The results showed that the surgical approach after conservative approaches has a lower recurrence rate than conservative approaches only (OR = 4.10, 95% CI = 1.38–12.23, P = 0.01). The total heterogeneity was not significant among the studies ($I^2$ = 28%, P = 0.25). Conservative approaches group has a recurrence rate of 36.25% and 10.93% for surgery after the conservative intervention group.

## Length of hospital stay

The length of hospital stay was reported in three studies [3, 6, 12], comparing the length of hospital stay after conservative treatment and surgery (Fig 4). In total, 1,113 patients were

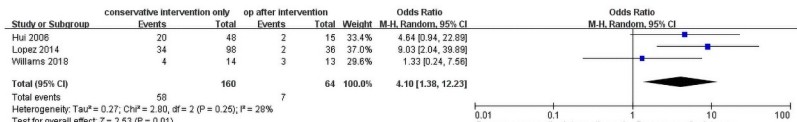

**Fig 3. Recurrence rate of conservative approaches only versus surgical approach after initial conservative approaches.**

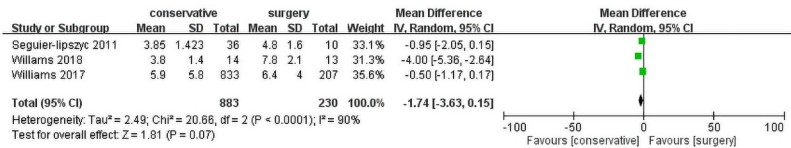

**Fig 4. Length of hospital stay.**

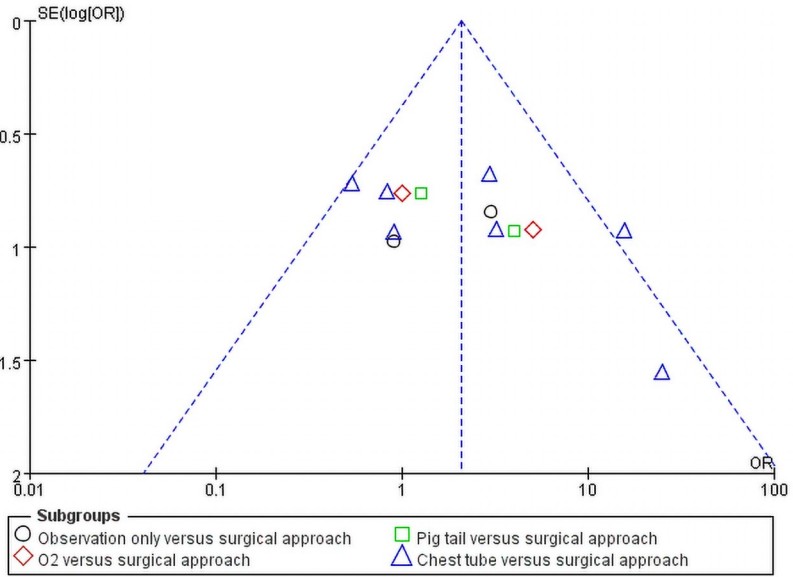

**Fig 5. Publication bias.**

included (883 and 230 patients underwent conservative invention and surgery, respectively). The forest plot revealed no significant differences (mean difference = −1.74, 95% CI = −3.63 to 0.15, P = 0.07). Heterogeneity was significant among the studies ($I^2$ = 90%; P < 0.0001).

## Publication bias detection

The outcomes are shown in Fig 5. In total, 13 subgroups were included. Studies were spread evenly on both sides of the average, indicating the absence of any publication bias.

## Discussion

To our knowledge, this is the first study to systematically review available articles that compare the effectiveness of different procedures as first-line management of PSP in the juvenile population. Considering our main outcome, our meta-analysis showed that as first-line management for PSP in young people, the surgical approach led to a significantly lower recurrence rate than did the conservative approaches.

Pneumothorax management has not been completely standardized, particularly for children. Supplemental oxygen use is based on limited evidence from small studies involving adult patients and animal models [19, 20]. A hypothesis states that the inhalation of a high concentration of oxygen might increase the absorption rate of gas from the pleural cavity. A small clinical study around the 1970s revealed that adults administered 100% supplemental oxygen therapy demonstrated a fourfold increase in the mean absorption rate [21]. In an observational study, PSP patients aged approximately 19 years benefited from oxygen therapy, which increased the PSP resolution rate [22]. However, for neonates, the use of 100% oxygen in pneumothorax treatment indicated little to no advantage compared with the use of room air alone [23].

Of all the PSP treatment methods, chest tube drainage and simple aspiration are the most popular first choices for adult patients with the first episode of PSP and were both applied in juvenile pneumothorax. Needle aspiration was preferred to tube drainage according to the British Thoracic Society (BTS) pleural disease guideline in 2010 [24] due to the equal outcomes

of the two procedures but shorter length of hospital stay after needle aspiration. However, American College of Chest Physicians emphasized chest tube drain as the first-line management because of the lower risk of persistent air leaking and additional procedures in chest tube drain than in needle aspiration [25]. In current practice, patients with PSP with respiratory distress receive needle aspiration or tube drainage first [13, 15, 26]. Surgery is indicated when persistent air leakage occurs [12–16, 26].

Because of a lack of sufficient pediatric data, PSP is managed according to adult BTS guidelines. Soccorso [7] performed a retrospective review, following BTS guidelines of PSP management in children. Notably, the results revealed that the guidelines are not applicable to children with large PSP and a relevantly high recurrence rate (36%) after the nonsurgical treatment.

Among studies we included in this meta-analysis, there was no major complication mentioned in the surgery intervention group (including VATS, thoracotomy, initial surgery and surgery after failed conservative procedure). The role of surgical intervention remains controversial. Most surgeons nowadays consider surgery after persistent air leaks or recurrent PSP. Whether immediate surgery ensures recurrence prevention remains uncertain. Because it has been widely applied in thoracic surgery, VATS is accepted as a safer and more feasible PSP management procedure compared with open thoracotomy[27–29], particularly in children with PSP [4, 5, 30]. Although some evidence has indicated VATS may be associated with recurrence risk [26, 30], it still plays an important role in juvenile PSP treatment. Furthermore, pneumothorax recurrence risk is not associated with surgical failure per se but with the potential new bulla formation [30]. The rapid growth in vertical rather than horizontal dimension can affect the pressure at the lung apex and prompt new bullae formation [31]. Weaver et al. [32] performed a morphometric analysis between rib cage and age, indicating that the overall size of the rib cage increased in three dimensions from 6 months to 20 years and only slightly increased from 20 to 30 years. The horizontal rib cage growth rate is faster than anterior–posterior and lateral dimensions before the age of 20 years. After age 30 years, the rib cage will gradually become round in shape to form a barrel-like structure. Interestingly, this result concurs with a review of Taiwanese PSP [8]. The PSP incidence and 1-year recurrence rate both decrease significantly once patients reach 20 years of age.

According to the 2010 adult BTS guideline, adult patients undergo different nonoperative approaches according to their clinical condition: disease stability and pneumothorax size [24]. Current practice of PSP management among the American Pediatric Surgical Association members was reviewed in 2019 [33], which showed a great variation in management between VATS timing. Moreover, surgeons prefer performing VATS if the initial conservative management fails [33]. However, in young patients, primary VATS can decrease the recurrence rate and even the length of hospital stay compared with conservative approaches [29, 34]. Williams et al. stated that in most pediatric patients, a definitive surgical management is required even after the initial conservative treatment [6, 12]. Lopez et al. found that the initial nonoperative management of patients with PSP resulted in longer lengths of hospital stay (median: 11 vs 5 days, P < 0.001). Moreover, more than one-third (37%) of the 108 patients after the initial nonoperative management eventually required VATS during their hospitalization [14].

Cook et al. reported the different costs involved in the initial surgical intervention for PSP and surgery for recurrence after nonoperative management in 15 patients. The costs were US $213,373 and US$229,576, respectively [35]. A single-center, evidence-based protocol was developed in 2019 for the management of pediatric patients with PSP. The average length of hospital stay decreased from 4.5 to 2.9 days. An early surgery was performed when nonsurgical methods failed or persistent air leak occurred for 2 days. Moreover, the overall cost decreased due to less radiology cost [36]. In this study, the length of hospital stay varied in our selected

studies [3, 12]. This could be due to variation in the observation duration before undergoing surgery. Some were operated after 3 days, but some waited for at least 5 days [12, 33]. As aforementioned, most young patients with PSP eventually need surgical management. Given the recurrence rate, financial expense, and length of hospital stay, surgical intervention might be considered an initial and feasible option for PSP management in children and adolescents.

Computed tomography scan is more sensitive than plain radiography for bleb detection because only 15% of the blebs are detected through plain chest radiography [37]. Furthermore, some studies have reported the size and number of blebs correlated with the likelihood of PSP recurrence [38–40]. Owing to a lack of relevant data in these selected studies, we could not analyze the correlation of bleb size and recurrence rate to determine the optimal approach. Miscia et al. [41] reported recurrence rate in children with bullae compared with those with no bullae detected based on a computed tomography scan. The recurrence rate is compatible between the groups, indicating that the presence of bullae noted in computed tomography images is not an appropriate parameter for PSP recurrence.

Regarding our main outcome, each subgroup of different conservative approaches and surgical approach showed similar recurrence rate, but only a few studies were included in our separated subgroups. When we analyzed the nonsurgical and surgical group recurrence rates as a whole, with increasing sample size, the group with surgery as the first-line treatment had a lower recurrence rate. Furthermore, in patients who underwent conservative treatment first and received surgery later, the recurrence rate was lower than in those who received nonsurgical treatment. Therefore, surgery might be a safe and sufficient first-line management intervention for juvenile PSP. Moreover, it is the optimal choice for patients with persistent air leakage or recurrent PSP after the nonsurgical treatment.

Several studies [3, 13, 18] included in this meta-analysis have selected mechanical pleurodesis as an adjunctive procedure. Mechanical pleurodesis was performed through mechanical abrasion of pleural surfaces or partial removal of parietal pleura, which creates adhesion between the two membrane layers in the pleural cavity [42]. Bialas et al. [43] demonstrated that mechanical and chemical pleurodesis with blebectomy have comparable outcomes. Moreover, a Taiwan study showed that pleurodesis reduces the recurrence rate and the need for further surgical intervention in pediatric PSP [44]. Although the ideal area and size of abrasion are still under debate, we expect better adhesion after extensive abrasion [45]. Some studies reported that the complication included postoperative dull chest pain, bleeding, or hemothorax [45, 46]. The postoperative complication may explain the prolonged length of hospital stay in our study's surgical approach group.

Huang et al. reported only 3.1% of PSP patients experienced recurrence more than four years after the initial occurrence and recommended five years as an adequate follow-up duration [8]. Some of our included studies performed in patient follow-up less than five years. Whereas most studies had follow-up duration around five years.

We, however, acknowledge the limitations of this study. Firstly, all the articles included in this research were retrospective, thereby resulting in more risk of bias. Secondly, some studies included Marfan syndrome or cystic fibrosis or patients with smoking habit with spontaneous pneumothorax as PSP. Thirdly, there is no consensus guidelines for management of PSP, the intervention selection is based on doctors' preference. With different PSP definitions, intervention strategies and follow up durations, there was significant risk of bias in some of the included studies and led to weaker strength of our recommendation. Fourthly, is the small sample size of our study. One study [6] we included had a large sample but was not included in our PSP recurrence rate analysis because authors did not perform primary surgery as their intervention. Rest of the studies in this meta-analysis reporting the primary outcome had <100 patients. The small sample size reduces the statistic power of this study. Finally, some

patients included in the studies were not first-episode PSP patients. Owing to a lack of medical history, we could not analyze treatment outcomes between PSP recurrence and first-episode PSP. Randomized controlled trials comparing surgical and nonsurgical management are needed in future to provide further evidence for juvenile-specific PSP guidelines.

## Conclusions

To our knowledge, this study is the first systematic review and meta-analysis comparing the surgical approach and conservative intervention as the initial management strategy for juvenile PSP. We found that the surgical approach as first-line management might lead to better recurrence prevention than the conservative procedure. Moreover, the surgical approach after failed conservative management leads to a low recurrence rate. In general, an upfront surgery might be considered an optimal strategy for juvenile PSP management.

## Supporting information

**S1 Table. Search strategy.**
(DOCX)

**S2 Table. Quality assessment.**
(DOCX)

**S1 Checklist. PRISMA 2009 checklist.**
(DOC)

## Author Contributions

**Conceptualization:** Fu-Huan Huang.

**Data curation:** Chun-Shan Hung, Fu-Huan Huang.

**Formal analysis:** Chun-Shan Hung, Fu-Huan Huang.

**Investigation:** Chun-Shan Hung, Ten-Fang Yang, Fu-Huan Huang.

**Methodology:** Yang-Ching Chen.

**Resources:** Ten-Fang Yang.

**Software:** Ten-Fang Yang.

**Supervision:** Fu-Huan Huang.

**Writing – original draft:** Chun-Shan Hung, Fu-Huan Huang.

**Writing – review & editing:** Chun-Shan Hung, Yang-Ching Chen, Fu-Huan Huang.

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
