## [Decision Letter · Decision Letter 0]

24 Feb 2021

PONE-D-21-03770

Systematic review and meta-analysis on juvenile primary spontaneous pneumothorax: conservative or surgical approach first?

PLOS ONE

Dear Dr. Huang,

Thank you for submitting your manuscript to PLOS ONE. After careful consideration, we feel that it has merit but does not fully meet PLOS ONE’s publication criteria as it currently stands. Therefore, we invite you to submit a revised version of the manuscript that addresses the points raised during the review process.

We look forward to receiving your revised manuscript.

Kind regards,

Christopher Cao

Academic Editor

PLOS ONE

Additional Editor Comments:

The authors set out to perform a systematic review of the literature to examine whether early surgical intervention is superior to conservative management for management of primary spontaneous pneumothorax (PSP) in juvenile patients (aged <21 years). The authors performed a meta-analysis of the included studies.

There are currently two major international guidelines on the management of PSP: the British Thoracic Society guidelines, last updated 2010, and the American College of Chest Physicians Delphi consensus statement, published in 2001. Both guidelines pertain to the management of PSP in adults, and these recommendations are frequently extrapolated to juvenile patients. Current guidelines suggest referral for potential surgical intervention in the following instances: 1) second ipsilateral pneumothorax; 2) first contralateral pneumothorax; 3) synchronous bilateral pneumothorax; 4) persistent air leak (>5-7 days); 5) professions at risk (divers, pilots); 6) pregnancy.

There are several options for conservative management, including observation alone, application of supplemental O₂ (100%), needle aspiration, and chest tube drainage.

This study aims to examine the current literature to determine the incidence rate of recurrent primary spontaneous pneumothorax following conservative management and surgical intervention. If surgical intervention greatly reduces the incidence of recurrence, then an argument could be made to offer surgical intervention in first episode PSP.

Strengths

• The study protocol was registered with prospectively with PROSPERO

• A comprehensive database search was performed according to the PRISMA guidelines. Nine studies were included in the final systematic review and meta-analysis.

• No language restrictions were employed in the search

• Two independent reviewers screened title/abstract and extracted the data

Major issues

1. There was significant risk of bias in the included studies, this was documented in the supplementary table (S1) but was not mentioned in the results or considered in the discussion when making a recommendation based on the results.

2. The background incidence of recurrence following conservative management and surgical intervention was not mentioned in the introduction and the calculated incidence of recurrence from the meta-analysis was not reported in the results. These two pieces of information are required for the reader to place the results in context.

3. (Lines 73-76) The search strategy needs to be reviewed and correctly documented; I doubt that the current strategy would yield useful results. I suspect the commas should be the OR operator, and OR should be AND.

4. (Table 1) The authors should document follow up. If the studies all have different lengths of follow up then their reported incidence of recurrence will likely change and therefore meta-analysis may not be appropriate.

5. There is no mention of the risk of major complication following surgical intervention. Any consideration of surgical intervention should weight the risk and benefit of the procedure. If the reduced risk of recurrence is equal to or exceeded by the risk of major morbidity then surgery should not recommended. There should be a brief mention of the risk of major complication in the discussion – this should be well documented in the literature.

6. The authors need to discuss the effect of the various biases present in the systematic review on the strength of their recommendation in the discussion section.

Minor issues

7. (Line 3) The authors have used the acronym ‘PSP’ without defining the term ‘primary spontaneous pneumothorax’

8. The eligibility criteria are not mentioned in the methods section of the abstract (juveniles aged <21)

9. (Lines 39-40) this sentence is grammatically incorrect, I think a word is missing e.g. “Furthermore, no evidence-based pediatric-specific guidelines for spontaneous pneumothorax management exist”

10. (Lines 45-46) these last two sentences could be better written as a single sentence.

11. (Lines 48-52) this statement should clearly define your Population, Intervention, Comparator, Outcome. Intervention and outcome are mentioned, the Population should be better defined (patients aged <21) and the outcomes need to be mentioned (incidence of recurrence and hospital length of stay).

12. (Lines 57-58) previous works don’t need to be cited as evidence of experience with systematic reviews.

13. (Line 104) the authors should mention that the secondary outcome: hospital length of stay was assessed, and state whether mean ± standard deviation or median and interquartile range was used for analysis.

14. (Table 1) It may be worth including gender in the table if reported in the initial studies. It may also be worth breaking down the number of included patients to state how many underwent conservative management and how many underwent surgical intervention.

15. (Line 157) the authors refer to four groups, there should only be two: intervention and comparator.

16. (Line 161-168) it is unclear to me what is meant by “surgical approach after conservative approaches”. Are these patients who failed conservative management due to prolonged air leak, or had recurrent pneumothorax after conservative management?

17. A limitation of the study not mentioned is the small sample size of the included studies. One study, Williams 2017, had a large sample but only reported hospital length of stay. All the studies reporting the primary outcome had <100 patients. The potential effect of this bias on the magnitude of the treatment effect should be discussed.

18. Forest plots need headings

Journal Requirements:

2. Please revise your PRISMA flow chart to ensure that you have included all of the reasons that full text articles were removed (listing how many were excluded for each reason).

3. Thank you for providing your search terms for your search. Please also provide the full electronic Boolean search strategy used to identify studies with all search terms and limits for at least one database. Please attach this as supplementary file.

4. Please reformat Table 1 for readability. Please also rename this table "Study Characteristics.

"NO. The funders had no role in study design, data collection and analysis, decision to publish, or preparation of the manuscript."

Reviewers' comments:

Reviewer's Responses to Questions

**Comments to the Author**

1. Is the manuscript technically sound, and do the data support the conclusions?

Reviewer #1: Partly

2. Has the statistical analysis been performed appropriately and rigorously? 

Reviewer #1: I Don't Know

3. Have the authors made all data underlying the findings in their manuscript fully available?

Reviewer #1: Yes

4. Is the manuscript presented in an intelligible fashion and written in standard English?

Reviewer #1: Yes

5. Review Comments to the Author

Reviewer #1: Summary of the research and overall impression

The authors set out to perform a systematic review of the literature to examine whether early surgical intervention is superior to conservative management for management of primary spontaneous pneumothorax (PSP) in juvenile patients (aged <21 years). The authors performed a meta-analysis of the included studies.

There are currently two major international guidelines on the management of PSP: the British Thoracic Society guidelines, last updated 2010, and the American College of Chest Physicians Delphi consensus statement, published in 2001. Both guidelines pertain to the management of PSP in adults, and these recommendations are frequently extrapolated to juvenile patients. Current guidelines suggest referral for potential surgical intervention in the following instances: 1) second ipsilateral pneumothorax; 2) first contralateral pneumothorax; 3) synchronous bilateral pneumothorax; 4) persistent air leak (>5-7 days); 5) professions at risk (divers, pilots); 6) pregnancy.

There are several options for conservative management, including observation alone, application of supplemental O₂ (100%), needle aspiration, and chest tube drainage.

This study aims to examine the current literature to determine the incidence rate of recurrent primary spontaneous pneumothorax following conservative management and surgical intervention. If surgical intervention greatly reduces the incidence of recurrence, then an argument could be made to offer surgical intervention in first episode PSP.

Strengths

• The study protocol was registered with prospectively with PROSPERO

• A comprehensive database search was performed according to the PRISMA guidelines. Nine studies were included in the final systematic review and meta-analysis.

• No language restrictions were employed in the search

• Two independent reviewers screened title/abstract and extracted the data

Major issues

1. There was significant risk of bias in the included studies, this was documented in the supplementary table (S1) but was not mentioned in the results or considered in the discussion when making a recommendation based on the results.

2. The background incidence of recurrence following conservative management and surgical intervention was not mentioned in the introduction and the calculated incidence of recurrence from the meta-analysis was not reported in the results. These two pieces of information are required for the reader to place the results in context.

3. (Lines 73-76) The search strategy needs to be reviewed and correctly documented; I doubt that the current strategy would yield useful results. I suspect the commas should be the OR operator, and OR should be AND.

4. (Table 1) The authors should document follow up. If the studies all have different lengths of follow up then their reported incidence of recurrence will likely change and therefore meta-analysis may not be appropriate.

5. There is no mention of the risk of major complication following surgical intervention. Any consideration of surgical intervention should weight the risk and benefit of the procedure. If the reduced risk of recurrence is equal to or exceeded by the risk of major morbidity then surgery should not recommended. There should be a brief mention of the risk of major complication in the discussion – this should be well documented in the literature.

6. The authors need to discuss the effect of the various biases present in the systematic review on the strength of their recommendation in the discussion section.

Minor issues

7. (Line 3) The authors have used the acronym ‘PSP’ without defining the term ‘primary spontaneous pneumothorax’

8. The eligibility criteria are not mentioned in the methods section of the abstract (juveniles aged <21)

9. (Lines 39-40) this sentence is grammatically incorrect; I think a word is missing e.g. “Furthermore, no evidence-based pediatric-specific guidelines for spontaneous pneumothorax management exist”

10. (Lines 45-46) these last two sentences could be better written as a single sentence.

11. (Lines 48-52) this statement should clearly define your Population, Intervention, Comparator, Outcome. Intervention and outcome are mentioned, the Population should be better defined (patients aged <21) and the outcomes need to be mentioned (incidence of recurrence and hospital length of stay).

12. (Lines 57-58) previous works don’t need to be cited as evidence of experience with systematic reviews.

13. (Line 104) the authors should mention that the secondary outcome: hospital length of stay was assessed, and state whether mean ± standard deviation or median and interquartile range was used for analysis.

14. (Table 1) It may be worth including gender in the table if reported in the initial studies. It may also be worth breaking down the number of included patients to state how many underwent conservative management and how many underwent surgical intervention.

15. (Line 157) the authors refer to four groups, there should only be two: intervention and comparator.

16. (Line 161-168) it is unclear to me what is meant by “surgical approach after conservative approaches”. Are these patients who failed conservative management due to prolonged air leak, or had recurrent pneumothorax after conservative management?

17. A limitation of the study not mentioned is the small sample size of the included studies. One study, Williams 2017, had a large sample but only reported hospital length of stay. All the studies reporting the primary outcome had <100 patients. The potential effect of this bias on the magnitude of the treatment effect should be discussed.

18. Forest plots need headings

Recommendation

Acceptable for publication with major alterations.

I would be happy to review a revised manuscript.

ADDIT:

My response to the question: Has the statistical analysis been performed appropriately and rigorously? I have selected 'I don't know' because it is currently unclear whether meta-analysis is appropriate. If, based on patient demographics and follow up, the studies are fairly homogenous then I would change my response to 'Yes'.

6. PLOS authors have the option to publish the peer review history of their article (what does this mean?). If published, this will include your full peer review and any attached files.

Reviewer #1: **Yes: **Nicholas McNamara

---

## [Author Response · Author response to Decision Letter 0]

28 Mar 2021

Thanks for your constructive suggestion; we have revised our manuscript as your suggestion and replied your question in " Response to reviewers".

---

## [Decision Letter · Decision Letter 1]

19 Apr 2021

Systematic review and meta-analysis on juvenile primary spontaneous pneumothorax: conservative or surgical approach first?

PONE-D-21-03770R1

Dear Dr. Huang,

We’re pleased to inform you that your manuscript has been judged scientifically suitable for publication and will be formally accepted for publication once it meets all outstanding technical requirements.

Kind regards,

Christopher Cao

Academic Editor

PLOS ONE

Additional Editor Comments (optional):

The authors have adequately addressed the issues raised by the Reviewer.

Reviewers' comments:

Reviewer's Responses to Questions

**Comments to the Author**

1. If the authors have adequately addressed your comments raised in a previous round of review and you feel that this manuscript is now acceptable for publication, you may indicate that here to bypass the “Comments to the Author” section, enter your conflict of interest statement in the “Confidential to Editor” section, and submit your "Accept" recommendation.

Reviewer #1: All comments have been addressed

2. Is the manuscript technically sound, and do the data support the conclusions?

Reviewer #1: Yes

3. Has the statistical analysis been performed appropriately and rigorously? 

Reviewer #1: Yes

4. Have the authors made all data underlying the findings in their manuscript fully available?

Reviewer #1: No

5. Is the manuscript presented in an intelligible fashion and written in standard English?

Reviewer #1: Yes

6. Review Comments to the Author

Reviewer #1: 1. The search strategy appears much more logical with the correct use of Boolean operators, however the placement of a few of the parentheses still appears incorrect.

2. (line 156) and (line 161-162) should say that there was no significant heterogeneity, rather than “heterogeneity showed no significant difference”. The wording of the results regarding heterogeneity is appropriate in all other paragraphs.

7. PLOS authors have the option to publish the peer review history of their article (what does this mean?). If published, this will include your full peer review and any attached files.

Reviewer #1: No

---

## [Editor Report · Acceptance letter]

21 Apr 2021

PONE-D-21-03770R1 

Systematic review and meta-analysis on juvenile primary spontaneous pneumothorax: conservative or surgical approach first? 

Dear Dr. Huang:

I'm pleased to inform you that your manuscript has been deemed suitable for publication in PLOS ONE. Congratulations! Your manuscript is now with our production department. 

Kind regards, 

on behalf of

Dr. Christopher Cao 

Academic Editor

PLOS ONE